# Synthesis of Gold Nanoparticles (AuNPs) Using *Ricinus communis* Leaf Ethanol Extract, Their Characterization, and Biological Applications

**DOI:** 10.3390/nano9050765

**Published:** 2019-05-18

**Authors:** Hamed A. Ghramh, Khalid Ali Khan, Essam H. Ibrahim, William N. Setzer

**Affiliations:** 1Research Center for Advanced Materials Science (RCAMS), King Khalid University, P.O. Box 9004, Abha 61413, Saudi Arabia; hamedsa@hotmail.com; 2Unit of Bee Research and Honey Production, Faculty of Science, King Khalid University, P.O. Box 9004, Abha 61413, Saudi Arabia; 3Biology Department, Faculty of Science, King Khalid University, P.O. Box 9004, Abha 61413, Saudi Arabia; essamebrahim@hotmail.com; 4Blood Products Quality Control and Research Department, National Organization for Research and Control of Biologicals, Cairo 12611, Egypt; 5Department of Chemistry, University of Alabama in Huntsville, Huntsville, AL 35899, USA; setzerw@uah.edu

**Keywords:** *Ricinus communis*, nanoparticles, chemical characterization, cancer cell lines, antimicrobial activity, hemolytic activity, cytotoxic effects

## Abstract

The purpose of this study was to explore the collective biological properties of *Ricinus communis* ethanol leaf extract (RcExt) and extract-fabricated gold nanoparticles (RcExt-AuNPs). AuNPs were synthesized using RcExt. Fingerprint data of the biochemicals putatively found in RcExt were obtained using gas chromatography–mass spectrometry (GC-MS/MS) and high-performance liquid chromatography/ultraviolet-visible (HPLC/UV-VIS) analyses. RcExt-AuNPs were characterized by UV-Vis spectroscopy, scanning electron microscopy (SEM), and Fourier- transform infrared radiation (FTIR) spectroscopy. Cytotoxic activity on the Hela and HepG2 tumor cell lines was tested through cell viability, antimicrobial activity against bacterial and fungal pathogens through a well diffusion assay, hemolytic activity on red blood cells through absorbance reading, and stimulatory/inhibitory effects on splenic cells by cell viability. AuNPs of 200 nm size were synthesized. GC-MS/MS analysis revealed 12 peaks and HPLC/UV-VIS analysis resulted in 18, 13, and five peaks at the wavelengths of 220, 254, and 300 nm, respectively. Cytotoxicity screening revealed that RcExt had stimulatory effects (6.08%) on Hela cells and an inhibitory effect (−28.33%) on HepG2 cells, whereas RcExt-AuNPs showed inhibitory effects (−58.64% and −42.74%) on Hela and HepG2 cells, respectively. Antimicrobial activity of RcExt-AuNPs against tested pathogens was significantly higher (average diameters of inhibition zones were higher (ranging from 9.33 mm to 16.33 mm)) than those of RcExt (ranging from 6.00 mm to 7.33 mm). RcExt and RcExt-AuNPs showed 4.15% and 100% lytic effects, respectively. Inhibitory effects on splenic cells for RcExt-AuNPs were observed to be significantly higher (−30.56% to −72.62%) than those of RcExt (−41.55% to −62.25%) between concentrations of 25 to 200 µg/mL. RcExt-AuNPs were inhibitory against HepG2 and Hela cells, while RcExt inhibited HepG2 but stimulated Hela cells. RcExt-AuNPs showed comparatively more antimicrobial activity. RcExt was safe while RcExt-AuNPs harmful to red blood cells (RBCs). RcExt and RcExt-AuNPs showed inhibitory effects on splenic cells irrespective of dose.

## 1. Introduction

*Ricinus communis*, with the common name castor oil plant, is a member of Euphorbiaceae. It is native to Africa but is currently found in each and every tropical region [1]. It has long been utilized in traditional Egyptian and Greek folk medicine and its uses have been defined in Ayurveda since the sixth century [2]. From root to fruit, all parts of the plant have medicinal importance. The fresh leaves are topically applied for headache and as a poultice for cysts and rheumatism; a decoction of the leaves used for emmenagogue. The outer layer of the root is laxative and effective in dermal illnesses. Castor seeds are rich in oil that stimulates the aperitive action and is recommended for the treatment of intestinal worms [1,2,3,4]. *R. communis* is used as a prophylactic, hypoglycemic, and purgative medication, and for treating cool and tumors in Tunisia [5]. Castor oil has demonstrated potential antimicrobial, cancer chemo preventive, cell reinforcement, and antidiabetic properties [6,7,8]. Anti-inflammatory activity [9], antidiabetic activity [7], and antimicrobial activity of root extracts [10] of this plant have been reported. In the Indian subcontinent, the leaf, root, and oil extracted from seeds of *R. communis* have been utilized to cure aggravation and liver concerns [11]. Phytochemicals present in the various plant parts are responsible for its medicinal properties. Many phytochemicals have been reported from this plant, and phenylalanine, ricinine, the *N*-demethyl and *O*-demethyl analogues of ricinine, and sucrose are important ones [12]. The flavonoids present in *R. communis* leaves include 3-*O*-xylosides, -glucosides, and -rutinosides of quercetin and kaempferol [13]. The major antioxidant phenolic compounds of *R. communis* leaves include quercetin, rutin, epicatechin, gentisic acid, gallic acid, and ellagic acid [14]. Lupeol and diketone pentacyclic triterpenes, from the *n*-hexane fractions of the root of *R. communis*, have exhibited significant anti-inflammatory activity [15].

Nanoparticles (NPs) fabricated with plant extracts are currently well-known and considered to be environmentally friendly approaches to prepare nanomaterials [16,17]. Phytochemicals, such as alkaloids, flavonoids, tannins, terpenoids, glycosides, and phenylpropanoids, present in plant extracts have been employed in these environmentally benign means of NP preparation. Plant extracts have latent biological activities, such as anti-inflammatory, antimicrobial, antioxidant, antimutagenic, or antihyperglycemic activities, and these properties may also manifest themselves in the biotic functions of the subsequent colloidal nanoparticle solution; therefore, plant extracts can be used in medicinal applications [18,19]. Numerous research findings have demonstrated that NPs ameliorated with natural products display biological activities [19]. Bioactive compounds in the plant extract take part in the reduction of metal ions that further function as capping agents for NPs and have shown bacteriostatic properties against Gram-positive and Gram-negative bacterial strains [20]. Many plant extract–NP bioconjugates have shown encouraging antioxidant activity and, thus, also have potential for development of novel therapeutic candidates against injuries caused by oxidative stress [16]. Plant extract–NP bioconjugates have displayed both in-vitro and in-vivo anti-inflammatory properties [21]. The wound-healing properties of plant extract–NP bioconjugates may be attributed to their antimicrobial, antioxidant, and anti-inflammatory activities. Furthermore, synthetic plant extract–NP nanomaterials may enhance the bioavailability and biocompatibility of the resulting NPs [19].

Recently, some work was done on the synthesis of zinc oxide NPs utilizing seeds of *R. communis*, and their biological activities, including antioxidant, antifungal, and anticancer activities, were investigated [22]. Nevertheless, there is still a big gap in the research to investigate the collective results of biological properties of the various metals with this plant extract. So, this study was designed to prepare gold nanoparticles (AuNPs) using the ethanolic leaf extract of *R. communis* (RcExt) with further investigation of the collective biological properties of both RcExt and its gold NPs (RcExt-AuNPs). Cytotoxic effects on the HepG2 and Hela cancer cell lines, antimicrobial activity against selected pathogens, hemolytic activity, and effect on splenic cell proliferation were biological properties that were investigated for both RcExt and RcExt-AuNPs.

## 2. Materials and Methods

### 2.1. Plant Extract Preparation

*R. communis* leaves were collected in September 2017 from Abha, Saudi Arabia. Plant identification was performed by a plant taxonomist at King Khalid University. Tap water was used to clean the leaves many times with the aim of eliminating dust and soil particles from their surfaces. Afterward, the leaves were cleaned with distilled water and desiccated in the dark at room temperature. Dry leaves were pulverized into powder using an electrical blender. To prepare the plant extract, a Soxhlet apparatus was used and 300 mL of 70% ethanol was added to 50 g of leaf powder. The resulting extract was filtered using Whatman filter paper (Merck, Darmstadt, Germany) and the solvent was evaporated under vacuum by rotary evaporator at 45 °C for 1–3 h to give about 2 g of crude extract in semi-solid state. One gram of this dry substrate was dispersed in 100 mL of 70% acetone to obtain a 1% stock solution.

### 2.2. Synthesis of AuNPs Using R. communis Leaf Extract

AuNPs were synthesized by following [23]. Briefly, 1 mL of 1 mM tetrachloroauric (III) acid trihydrate (HAuCl_4_·3H_2_O) solution was mixed with 99 mL of 1% stock solutions of RcExt. The solution was kept at room temperature and stirring was continued until its color was transformed to a golden-brown representing the development of nanoparticles.

### 2.3. Characterization of AuNPs Fabricated with R. communis Leaf Extract

NPs were subjected to UV/Vis spectra (λ range = 300 nm to 700 nm) in a UV-3600 Shimadzu spectrophotometer (Shimadzu Corporation, Kyoto, Japan) with an interpretation of 1 nm according to [24]. Their shape was described through a scanning electron microscope (SEM, JEM-1011, JEOL, Tokyo, Japan) with an accelerating voltage of 90 KV. The functional groups present in RcExt-AuNPs were examined by Perkin-Elmer Spectrum 2000 FTIR (PerkinElmer Inc., Waltham, MA, USA) between the range of 0–4000 cm^−1^ @ 16× and the precision of 4 cm^−1^. An X-ray diffraction (XRD) measurement of the RcExt-AuNPs was recorded with 2θ value in the range of 20 to 80° with a scan rate of 1° min^−1^ on fine layers of the corresponding liquid drops coated on a microscopic glass slide using an X-ray diffractometer (Rigaku Cooperation, Tokyo, Japan) operated at 40 KV and 30 mA with Cu Kα1 radiation.

### 2.4. Gas Chromatography–Mass Spectrometry Analysis of RcExt

*R. communis* leaf extract was analyzed by GC-MS/MS using a Gas Chromatograph with a Thermo Scientific (Austin, TX, USA) TSQ 8000 Triple Quadrupole mass selective detector (operated in the electron ionization (EI) mode, scan range = 45–650 amu, scan rate = 3.99 scans/s), and a Thermo Xcaliber. The GC column was TG-5SLIMS, with a capillary film thickness of 0.25 µm, a length of 30 m, and an internal diameter of 0.25 mm. The carrier gas was helium with a column head pressure of 48.7 kPa and a flow rate of 1.0 mL/min. The inlet temperature was 220 °C and the interface temperature was 280 °C. The following GC oven temperature program was used: 80 °C initial temperature, hold for 1 min; increased at 35 °C/min to 185 °C; increased 5 °C/min to 240 °C; increased 10 °C/min to 300 °C. A 0.5% w/v solution of RcExt in methanol was prepared and 1 μL was injected using a split-less injection technique. Identification of the components was based on their retention indices determined by comparison of their mass spectral fragmentation patterns with those stored in the MS library (NIST MS Search 2.0). The percentages of each component are reported as raw percentages based on total ion current without standardization.

### 2.5. High-Performance Liquid Chromatography/Ultraviolet-Visible (HPLC/UV-VIS) Analysis of RcExt

Chromatographic experiments were conducted on a Shimadzu (Shimadzu Corporation, Kyoto, Japan) HPLC instrument comprising quaternary LC-10A VP pumps, a variable wavelength programmable UV-visible detector, an SPD-10A VP column oven (Shimadzu Corporation, Kyoto, Japan), and a SCL 10A VP system controller. The instrument was controlled by use of Class VP 5.032 software (Shimadzu Corporation, Kyoto, Japan) installed with the equipment. Dried plant extract (10 mg) obtained by a rotatory evaporator was dissolved in 10 mL of HPLC grade methanol and sonicated for dissolving. This sample solution was filtered through a 0.22 µL syringe and then injected using a Rheodyne injector fitted with a 20 µL fixed loop. The separation was achieved using a column with dimensions of 15 × 4.6 mm, a particle size of 5 µm, and a C18 reverse phase column. The mobile phase used consisted of acetonitrile and water in the ratio of 50:50 v/v. All the analyses were performed at room temperature and chromatograms were monitored at wavelengths of 220, 254, and 300 nm using a UV visible detector (Shimadzu Corporation, Kyoto, Japan).

### 2.6. Antimicrobial Activities of RcExt and RcExt-AuNPs

#### 2.6.1. Micro-Organisms and Media

The micro-organisms used to study the antimicrobial activity were provided by the Microbiological Laboratory, Faculty of Science, King Khalid University, Saudi Arabia: Gram-negative *Escherichia coli*, *Proteus mirabilis*, and *Shigella flexneri*; and Gram-positive *Staphylococcus aureus*. The fungus used in the experiment was *Candida albicans*. The bacteria and the fungus were handled through standard techniques as described by [25] and preserved at 4 °C on nutrient agar slants. Both nutrient agar and broth (HiMedia Laboratories Pvt. Ltd. Mumbai, India) were prepared as per the manufacturer’s instructions.

#### 2.6.2. Well Diffusion Method for Antimicrobial Activity

Antimicrobial potentials of RcExt and RcExt-AuNPs were determined by the agar well diffusion method. All the microbes were inoculated in 10 mL of broth and kept in an incubator (Sheldon Manufacturing, Inc. Cornelius, OR, USA) at 37 °C overnight. Agar plates were prepared in a laminar flow hood by pouring the autoclaved nutrient agar onto plates. The surfaces of the agar plates were inoculated by spreading 50 µL of each microbial suspension (~10^8^ colony forming units (cfu)/mL) with a sterile cotton swab (Citotest Labware manufacturing Co. Ltd., Haimen, China). Four holes with a diameter of 6 mm were punched aseptically with a sterile cork borer in each agar plate. Thirty microliters of RcExt and RcExt-AuNPs were introduced into the wells on the agar plates. HAuCl_4_·3H_2_O (1 mM) and Penicillin–Streptomycin (20 units:20 µg solution) were used as controls. All of the petri plates were kept in an incubator (Nüve Sanayi Malzemeleri, Ankara, Turkey) at 33 °C for 24 h. The diameter of the inhibition zone was determined in triplicate.

### 2.7. Splenic Cell Culture Preparation

A single-cell splenic suspension was prepared from rats according to [26]. Briefly, the spleen of an anesthetized healthy adult male Sprague Dawley rat (200 g), kindly supplied by the animal facility of King Khalid University, was homogenized to release splenocytes and single-cell suspensions were prepared in DMEM high-glucose culture medium containing 9% fetal calf serum. Cells were adjusted to 0.5 × 10^6^/mL. The cell culture was accomplished in a micro well tissue culture plate (Corning Life Sciences, Oneonta, NY, USA) with addition of 100 μL of cell suspension (5000 cells/well) and 100 μL of each of RcExt and RcExt-AuNPs at 25, 50, 100, and 200 μg/mL final concentrations. Plates containing the cell cultures were put in a humid chamber at 37 °C for 72 h in 5% CO_2_ (Thermo Fisher Scientific, Waltham, MA, USA) [27]. A Vybrant^®^ MTT Cell Proliferation Assay Kit (Thermo Fisher Scientific) was used, following the manufacturer’s instructions, to calorimetrically analyze the change in cell numbers of the treated cells [28]. The findings were denoted by an increase or a decrease in growth percentage by following [29].

### 2.8. Lytic Effects of RcExt and RcExt-AuNPs on Red Blood Cells (RBCs)

The lytic effects of RcExt and RcExt-AuNPs were determined by following the methodology of [29] with a few modifications. RcExt and RcExt-AuNPs (1 mg/mL) were prepared in sterilized phosphate-buffered saline (PBS). Cow’s blood (10 mL) in 15 mL Falcon tubes was gently mixed and centrifuged for 15 min at 1000× *g*. The resulting supernatant was poured off and RBCs were elucidated three times with PBS. RBCs were suspended in PBS to obtain hematocrit (10%). For each assay, 100 µL of prepared RcExt and RcExt-AuNPs stock solutions was mixed with 900 µL of hematocrit in Eppendorf tubes and left in an incubator for 45 min at 37 °C. Triton X-100 (1%) and PBS alone were used as positive and negative controls, respectively. Centrifugation of all Eppendorf tubes was carried out at 2000 rpm for 10 min. Once the centrifugation was complete, the absorbance of supernatants was measured at 576 nm (Lamda 25- Perkin Elmer Inc. Waltham, MA, USA). The experiments were repeated to obtain three biological replicates.

### 2.9. Effects of RcExt and RcExt-AuNPs on the Hela and HepG2 Cancer Cell Lines

HepG2 and HeLa cells (Merck, Darmstadt, Germany) were cultured separately in DMEM (Merck, Darmstadt, Germany) with addition of 10% fetal calf serum (Web Scientific, Cheshire, UK), penicillin/streptomycin (100 U/mL/100 mg/mL, (Web Scientific, Cheshire, UK), and 2 mM L-glutamine (Web Scientific, Cheshire, UK), at a cell density of 5000 cells/well in 96-well tissue culture plates (Corning Life Sciences, Oneonta, NY, USA). Plates were kept in an incubator at 37 °C under 5% CO_2_ and 90% relative humidity (RH) for 24 h. The media were removed from each plate and replaced with 200 µL/well fresh media containing 100 µg/mL RcExt and RcExt-AuNPs separately. Cells with media only served as a control culture. The cultures were continued for an additional 24 h under the same conditions described above. A cell viability assay was done according to [26] using a Vybrant^®^ MTT Cell Proliferation Assay Kit (Thermo Fisher Scientific) with minor modifications. The media in micro well plates were changed with 100 µL of new culture medium and 10 µL of 12 mM MTT was added to each well and left for 3 h. Then, to each well, 100 µL of 0.1% acidified sodium dodecyl sulfate (SDS) was added and an absorbance reading was taken at 570 nm. Results are displayed as the % age of the control at the completion of each incubation period.

### 2.10. Statistical Analysis

The average diameters of the zone of interest (ZOI) were measured from three replicates along with the standard deviation (SD). An analysis of variance (ANOVA) was performed through Statistix 8.1 software. A pairwise assessment to compare all the means was achieved with Tukey’s Honest Significant Difference (HSD) test. Means that differed at *p* ≤ 0.05 were measured as statistically significant. In the case of biological activities, the data were the means of three replicates. Variances in concentrations of RcExt and RcExt-AuNPs were evaluated by ANOVA by applying SPSS (version 17). Differences with *p* ≤ 0.05 were considered to be statistically significant.

## 3. Results

### 3.1. Characterization of RcExt-AuNPs

#### 3.1.1. Change in Color

The AuNPs synthesis was initially confirmed through the change in color of the reaction mixture from dark yellowish green to dark brown (Figure 1).

#### 3.1.2. UV-Vis Spectrometry

RcExt-AuNPs were analyzed by UV-visible spectroscopy. When the RcExt was mixed with an aqueous solution of HAuCl_4_·3H_2_O, UV-Vis spectra of the plant leaf extract did not show any indication of absorption between 520 and 570 nm. However, when the AuNPs were synthesized, an absorption peak at 536 nm was obvious, indicating the formation of nanoparticles (Figure 2).

#### 3.1.3. Morphological Characterization Using Scanning Electron Microscopy (SEM)

The size and shape of gold nanoparticles (Figure 3) were characterized through SEM. The analyses clearly showed the uniform spherical shape and low dimensions of AuNPs. It can be observed from the figure that the size of NPs is around 40–80 nm by keeping interparticle spaces, which were enlarged at ×30,000 times.

#### 3.1.4. FT-IR of AuNPs Phytofabricated by RcExt

The FT-IR spectra of RcExt-AuNPs are shown in Figure 4. The spectra show peaks at 3019, 2914, 2843, 1601, 1491, 1540, 1025, 745, 693, and 536 cm^−1^. The synthesis of gold nanoparticles resulted in three main peaks at 3019, 2914, and 2843 cm^−1^ because of the occurrence of bonded O–H stretching of alcohols/phenols and N–H vibration of amines. The medium band at 1601 cm^−1^ corresponded to conjugation effects of C=O stretching of carbonyl groups. The strong band 1025 cm^–1^ corresponded to aliphatic amines. The bands at 745 and 693 cm^–1^ corresponded to stretching of haloalkanes.

#### 3.1.5. XRD Analysis of RcExt-AuNPs

The conformation of the prepared AuNPs was investigated by the XRD technique, and the corresponding XRD patterns are shown in Figure 5. Gold nanocrystals exhibited four distinct peaks at 2θ = 38.1, 44.3, 64.5, and 77.7. All four peaks corresponded to standard Bragg reflections (111), (200), (220), and (311) of the face center cubic (fcc) lattice plan according to -JCPDS No. 04-0784.

#### 3.1.6. GC-MS/MS Analysis of RcExt

The composition of RcExt was determined by GC-MS/MS and the major components were identified (Table 1, Figure 6).

#### 3.1.7. HPLC/UV-VIS Analysis of RcExt

The mobile phase acetonitrile:water (50:50) at the wavelengths of 220, 254, and 300 nm resulted in 18, 13, and five peaks, respectively (Table 2, Table 3 and Table 4). HPLC/UV-VIS chromatograms of RcExt monitored at the wavelengths of 220, 254, and 300 nm are shown in Figure 7, Figure 8 and Figure 9.

### 3.2. Antimicrobial Activity

The antimicrobial activity (in vitro) of RcExt and RcExt-AuNPs against Gram-positive *S. aureus*, Gram-negative *E. coli*, *P. mirabilis*, and *S. flexneri*, and the fungus *C. albicans* is summarized in Figure 10. There were significant differences in antimicrobial activity (AMA) among RcExt and RcExt-AuNPs against all tested microbial strains. The AMA of HAuCl_4_·3H_2_O was statistically (*p* ≤ 0.05) comparable with (i) RcExt against *P. mirabilis* and *C. albicans* (ii) RcExt-AuNPs against *S. aureus*, but significantly (*p* ≤ 0.05) different from RcExt and RcExt-AuNPs against *E. coli* and *S. flexneri*. The AMA of the positive control was significantly (*p* ≤ 0.05) greater than that of HAuCl_4_·3H_2_O, RcExt, and RcExt-AuNPs against all studied microbial strains.

### 3.3. Cytotoxic/Proliferative Effects on Rat Splenic Cell Proliferation

The cytotoxic or stimulatory properties that may be found in the RcExt or RcExt-AuNPs were tested on normal rat splenic cells at various concentrations. The results showed that there were significant (*p* ≤ 0.05) inhibitory effects for both RcExt and RcExt-AuNPs (Figure 11). These inhibitory effects were not dose-dependent, but rather fluctuated with increasing extract concentrations.

### 3.4. Lytic Effects of RcExt and RcExt-AuNPs on RBCs 

The % age of RBC lysis was determined by comparing the absorbance of the specimen to the positive and negative controls (Table 5). Significant (*p* ≤ 0.05) differences were observed in hemolytic activity and absorbance among the tested samples. One hundred percent lysis was displayed by the positive control (1.5% Triton X-100), while the negative one (PBS) revealed no lysis to the RBCs. RcExt-AuNPs showed 100% lysis, while RcExt-AuNPs showed only 4.15% RBC lysis (Table 5 and Figure 12).

### 3.5. Effects of RcExt and RcExt-AuNPs on the Hela and HepG2 Cancer Cell Lines

RcExt alone showed a very low stimulatory effect on Hela cells and a significant (*p* ≤ 0.05) inhibitory effect on HepG2 cells (Figure 13). RcExt-AuNPs showed significant (*p* ≤ 0.05) inhibitory effects on both Hela and HepG2 cell lines.

## 4. Discussion

HPLC and GC-MS/MS analyses of RcExt were performed to obtain its possible chemical composition, particularly, to have an indication of whether it contains phenolics, alkaloids, and flavonoids as possible contributors to the biological activities of the extract. The obtained fingerprint data revealed the presence of UV active biological compounds, such as alkaloids and flavonoids. The HPLC/UV-Vis data can be used as an identification tool for UV active biological compounds. The synthesis of NPs was initially monitored and confirmed visually through color variation. The distinctive light-gold color RcExt after addition of HAuCl_4_·3H_2_O can be credited to the excitation of the surface plasmon response (SPR) of NPs as confirmed by [30]. Some chemical compounds (alkaloids, flavonoids, saponins, and steroids) may alter the color of the solution as they act as reducing agents with the help of reductase enzymes. These specific enzymes discharged into a mixture can reduce the HAuCl4·3H2O to AuNPs by capping agents, such as proteins [31]. UV-Vis spectroscopy is one of the best and extensively used methods for the structural characterization of NPs [32]. Bioreduction of gold ions by biomolecules in plant leaf extracts are the likely reason for this observation [33,34]. FT-IR spectra with an analysis of vibrations is a suitable means for measuring minor structures in interactions between metallic nanoparticles and biomolecules [35]. Figure 4 showed that the N–H bonding vibration of leading amines and C–N elongation and coincidence of aliphatic amines have the robust capability to bind metal; hence, the phytochemicals from the plant extract putatively make a coating that shields the metal nanoparticles. This suggests that plant metabolites may probably achieve the task of the creation and stabilization of the AuNPs. The precise mechanism important for the reduction of metal ions still needs to be clarified for *R. communis*. The particular shape and size of synthesized AuNPs can be examined by computing the XRD. The XRD spectrum measured in this study resulted in four intense peaks observed in the spectrum, which agreed with the Bragg reflection of AuNPs.

It is obvious from the results that RcExt and RcExt-AuNPs have significant antibacterial and antifungal activities against all examined microbial strains. Although the antimicrobial activities of gold nanoparticles have been previously reported, their precise mechanisms of activity are still poorly understood. Several mechanisms for the antimicrobial activities of silver and gold nanoparticles have been proposed. The antimicrobial activity of nanoparticles may be due to their small size and uniform distribution. The ultra-small size of nanoparticles is responsible for deep penetration and the diffusion of nanoparticles through the cell membranes of microbial pathogens to disrupt the cell function. The nanoparticles may bind to the surface of the micro-organisms and cause pits in the cell wall, resulting in the leakage of cell contents that leads to cell death [36]. In addition, the metal ions may also produce free radicals, which could also lead to cell membrane rupture and premature cell death [37]. Antimicrobial activity is boosted by reactive oxygen species (ROS) generation and surface potential [38]. The nanoparticles could also inhibit DNA replication by binding to the phosphate moieties of DNA [39]. In this study, an increased effect was observed in the AMA of RcExt-AuNPs (resulting from the reaction of RcExt and HAuCl_4_·3H_2_O) against all microbial strains except *S. flexneri*. Similar effects were also observed by [40] during a comparative antimicrobial and toxicological study of gold and silver complexes with aromatic heterocycles.

In this study, cell viability and proliferation was analyzed by the MTT assay. MTT is transformed to unsolvable tinted formazan crystals when it comes in contact with live and metabolically active cells. The quantity of resulting formazan is directly proportional to the number of cells [28]. The obtained result in this experiment showed that both RcExt and RcExt-AuNPs had inhibitory effects on splenocytes irrespective of dose. The compound ingenol mebutate found in *R. communis* is a growth inhibitory mediator [41,42] and could be the possible reason for this inhibitory effect.

Biocompatibility and biological safety are the primary concerns for any material to be used in cellular systems. Exogenous materials interact with various cellular systems and may lead to cell damage. The purpose of measuring hemolytic activity was to evaluate the biological safety of the studied materials. RcExt showed non-significant hemolysis. The results for RcExt hemolysis are in accordance with [43]; they tested the hemolytic activity of 11 plant extracts used in folk medicine. Ten of the 11 plants in their study did not show hemolytic activity. In our results, RcExt-AuNPs was 100% hemolytic. The lysis effect of extract-containing metal nanoparticles may be due to the direct effect of the nanoparticles on the membranes of the RBCs. Other investigators have suggested that the hemolytic effects are due to the induced release of oxidative stress products following exposure [44]. A previous study showed that *R. communis* extract treatment resulted in increased lactate dehydrogenase (LDH) leakage, DNA fragmentation, and apoptotic cell percentage as well as the production of reactive oxygen species (ROS) and a downregulation of intracellular glutathione levels. *R. communis* extract-actuated cell death was facilitated by the production of ROS, the subsequent initiation of the caspase-3 cascade, and downstream actions that ultimately resulted in apoptosis [45].

## 5. Conclusions

In conclusion, all characterization methods applied in the present study confirmed the synthesis of AuNPs. RcExt chemical profiling revealed the UV active biological compounds putatively responsible for biological activities. RcExt-AuNPs showed inhibitory effects on both the Hela and HepG2 cell lines, while the EpExt inhibited HepG2 but stimulated Hela cells. EpExt-AuNPs had more antimicrobial effects on the tested microbial strains than EpExt. RcExt was found to be safe, while RcExt-AuNPs were 100% hemolytic. Both EpExt and EpExt-AuNPs showed inhibitory effects on splenic cells irrespective of the dose. Further studies to isolate and characterize the active compound responsible for biological activities of this plant extract are recommended and may lead to the discovery of novel compounds and replace the conventional chemotherapeutic and antimicrobial agents.

## Figures and Tables

**Figure 1 nanomaterials-09-00765-f001:**
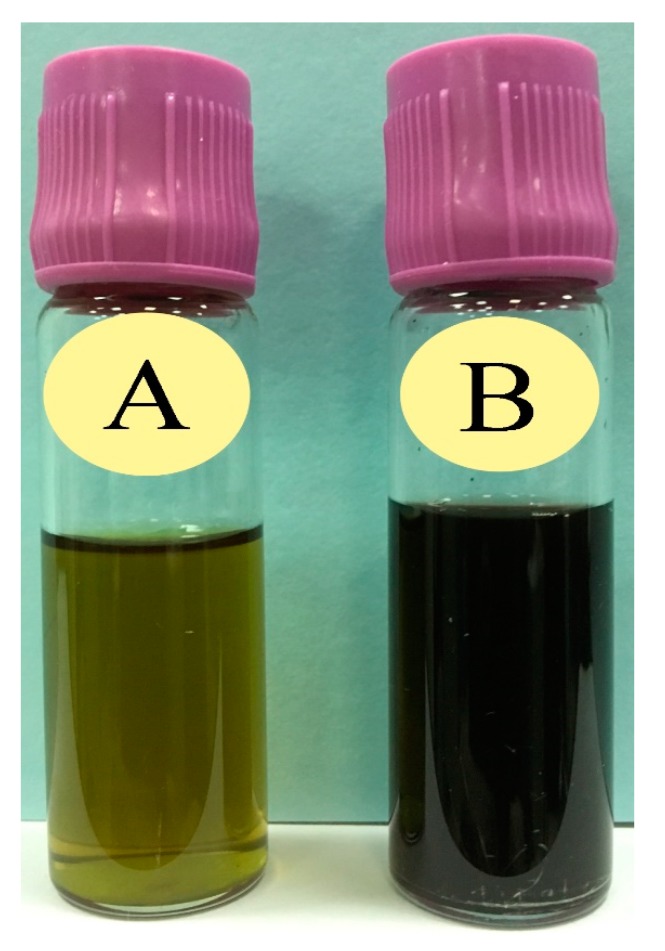
Confirmation of gold nanoparticles (AuNPs) synthesis through the change in color: (**A**) Prior to the addition of HAuCl4·3H2O in *Ricinus communis* leaf ethanol extract (RcExt) and (**B**) after the addition of HAuCl_4_·3H_2_O to RcExt.

**Figure 2 nanomaterials-09-00765-f002:**
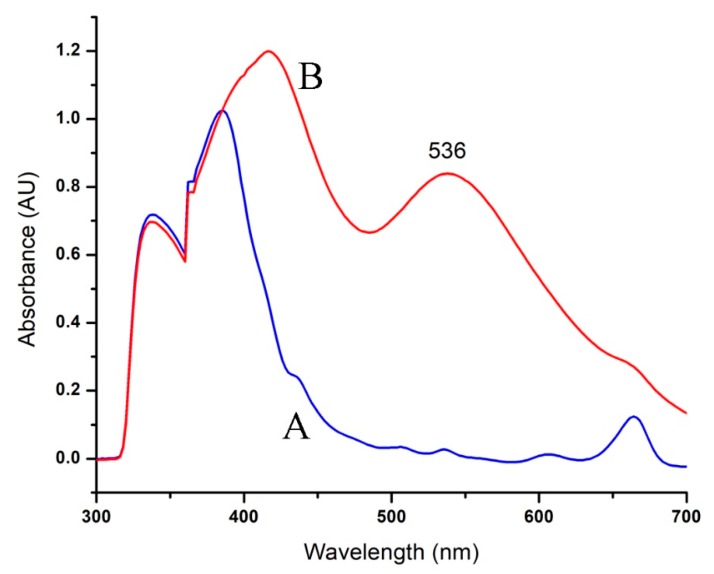
UV-Vis spectra of AuNPs formed through *R. communis* leaf extract (RcExt): (**A**) denotes the RcExt and (**B**) indicates the RcExt reacted with 1 mM solution of HAuCl_4_·3H_2_O.

**Figure 3 nanomaterials-09-00765-f003:**
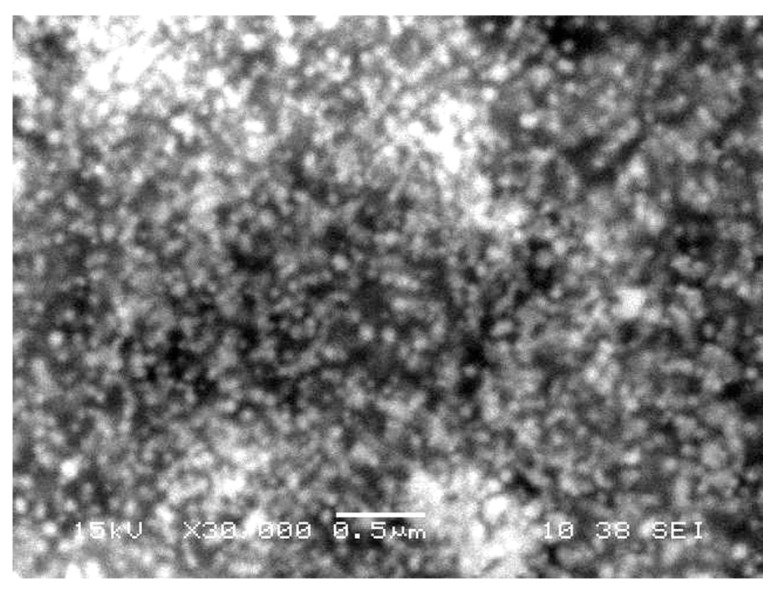
A scanning electron microscope image of gold nanoparticles phytofabricated by *Ricinus communis* leaf extract. The AuNPs are shown as round, crystalline, uniform aggregates.

**Figure 4 nanomaterials-09-00765-f004:**
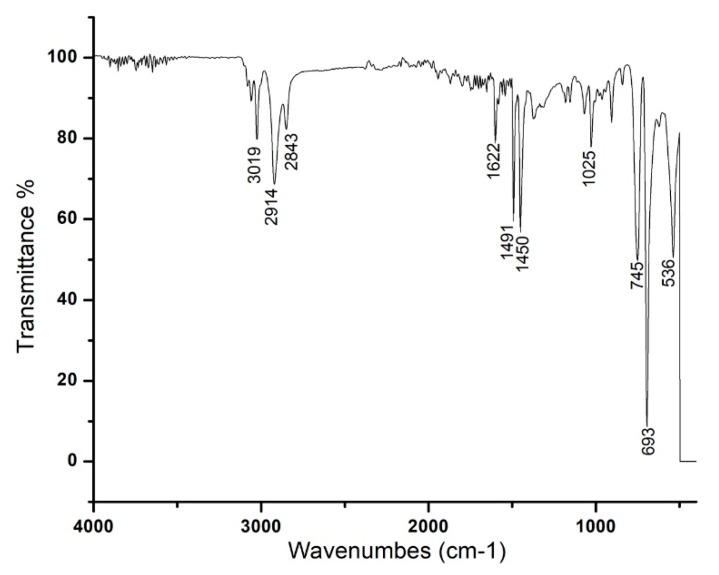
FT-IR spectra of the AuNPs phytofabricated by *Ricinus communis* leaf extract reduced with 1 mM solution of HAuCl_4_·3H_2_O.

**Figure 5 nanomaterials-09-00765-f005:**
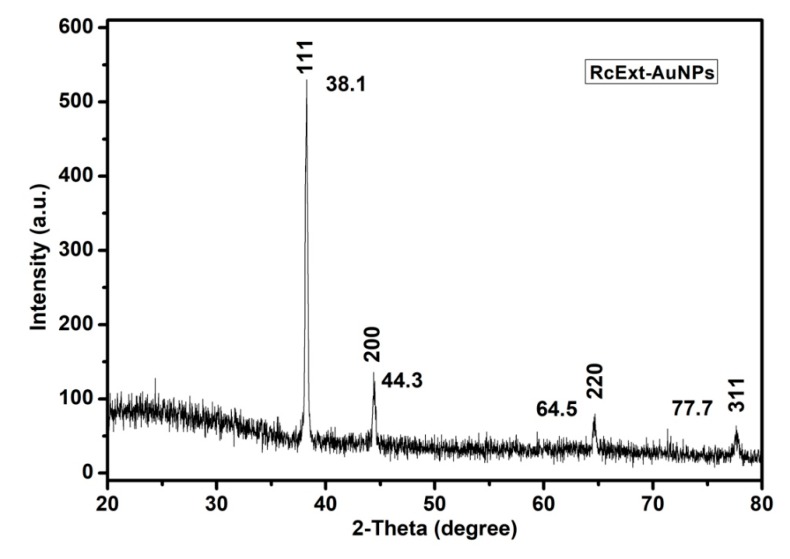
An X-ray diffraction (XRD) pattern of gold nanoparticles (AuNPs) obtained by using *Ricinus communis* ethanol leaf extract.

**Figure 6 nanomaterials-09-00765-f006:**
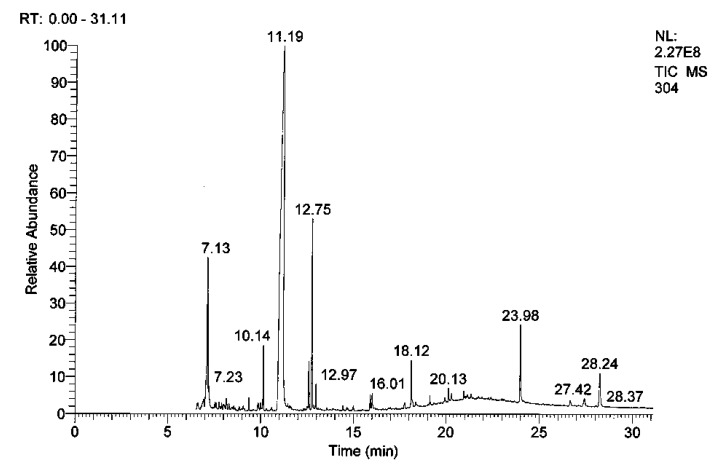
GC-MS/MS chromatogram of *Ricinus communis* leaf extract (RcExt).

**Figure 7 nanomaterials-09-00765-f007:**
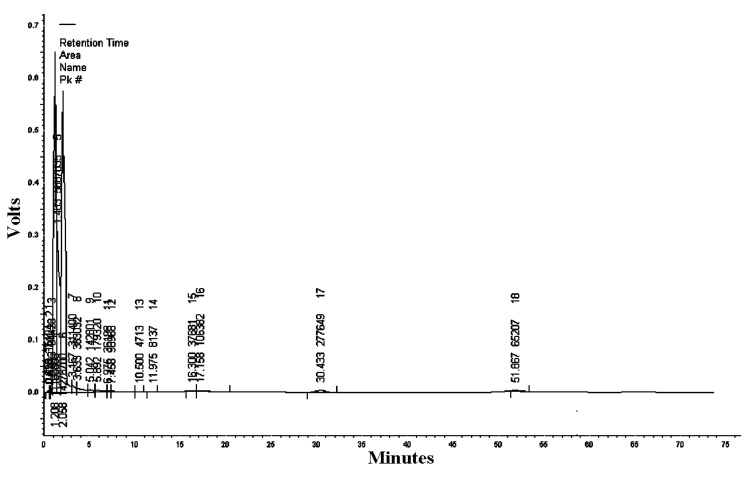
High-performance liquid chromatography/ultraviolet-visible (HPLC/UV-VIS) chromatogram of *Ricinus communis* leaf ethanol extract monitored at the wavelength of 220 nm.

**Figure 8 nanomaterials-09-00765-f008:**
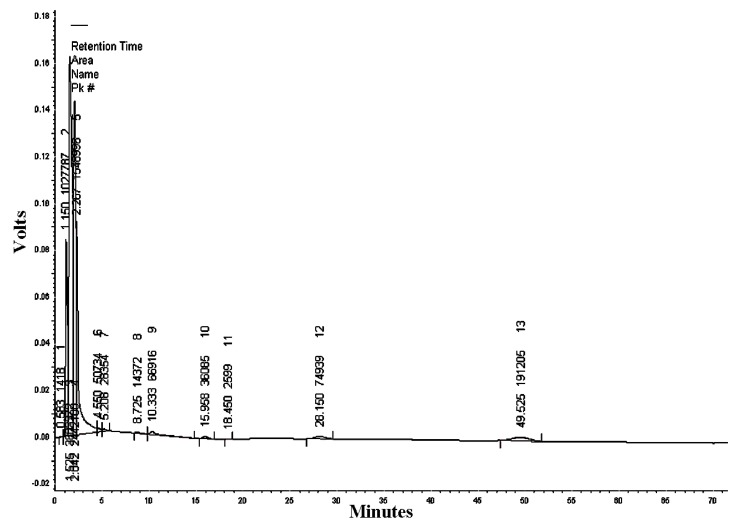
High-performance liquid chromatography/ultraviolet-visible (HPLC/UV-VIS) chromatogram of *Ricinus communis* leaf ethanol extract monitored at the wavelength of 254 nm.

**Figure 9 nanomaterials-09-00765-f009:**
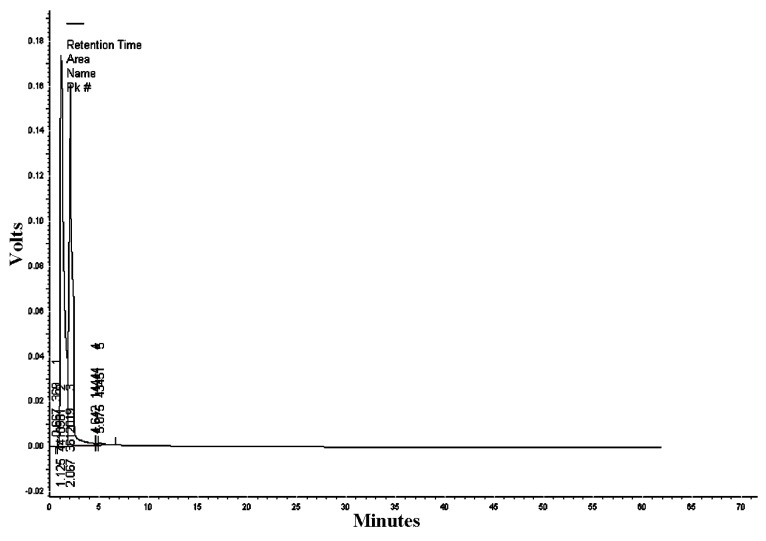
High-performance liquid chromatography/ultraviolet-visible (HPLC/UV-VIS) chromatogram of *Ricinus communis* leaf ethanol extract monitored at the wavelength of 300 nm.

**Figure 10 nanomaterials-09-00765-f010:**
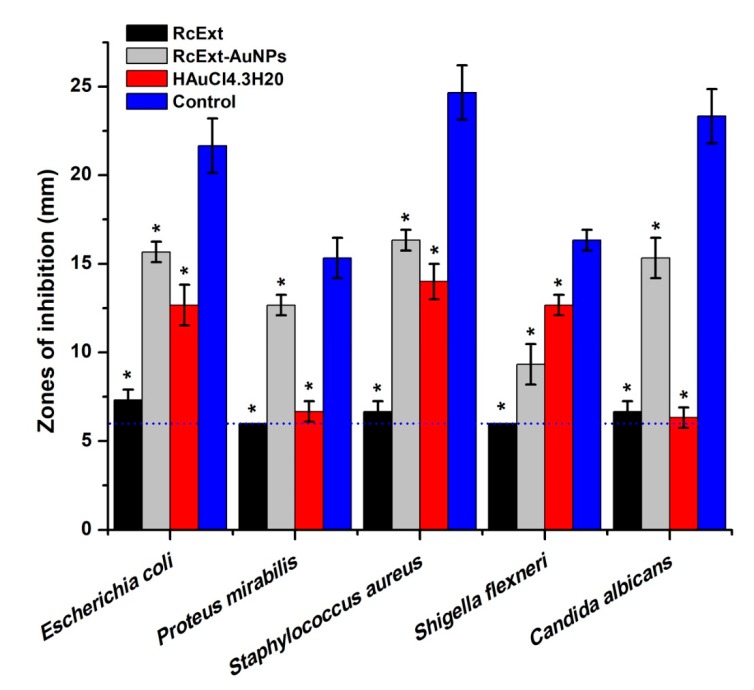
Antimicrobial potentials of (i) *Ricinus communis* leaf extract (RcExt); (ii) the extract with gold nanoparticles (RcExt-AuNPs); (iii) HAuCl_4_·3H_2_O (1 mM); and (iv) the control (Penicillin–Streptomycin (20 units) against selected microbial strains. Values ≤6.00 mm (the dotted line) imply no antimicrobial activity as it corresponds to the diameter of a well. All data are represented as mean ± standard deviation (indicated as error bars) from three determinations. Means with asterisk (*) signs in each microbial group indicate a significant (*p* ≤ 0.05) difference among the treatments and control.

**Figure 11 nanomaterials-09-00765-f011:**
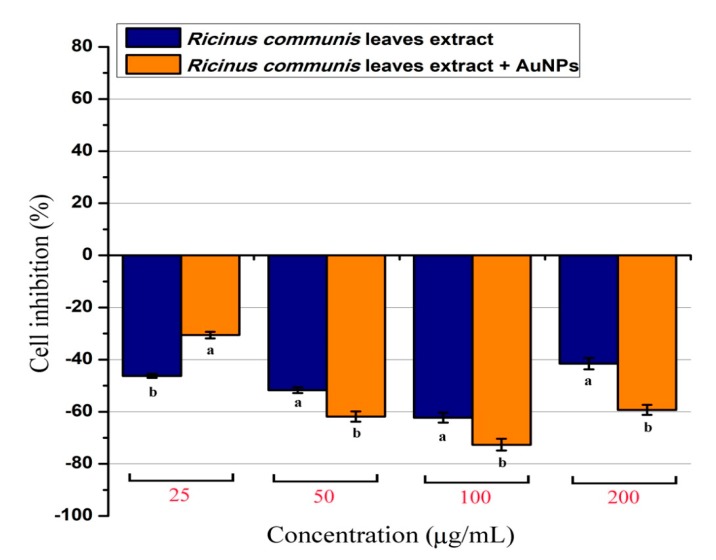
Percentage of normal splenic cells growth stimulation/inhibition after treatment with *Ricinus communis* leaf extract (RcExt) and gold nanoparticles fabricated with extract (RcExt-AuNPs). All of the data are represented as mean ± standard deviation (indicated as error bars) from three determinations. Bars with letters in each concentration group indicate a significant (*p* ≤ 0.05) difference between treatments.

**Figure 12 nanomaterials-09-00765-f012:**
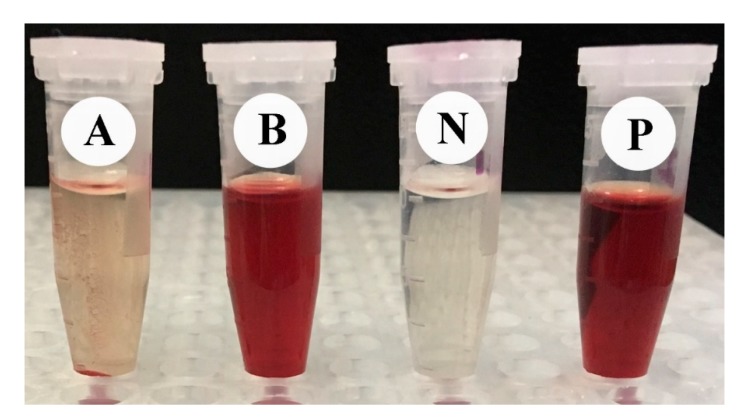
The hemolytic effect of (**A**) *Ricinus communis* ethanol leaf extract (RcExt) and (**B**) extract with gold nanoparticles (RcExt-AuNPs) on RBCs. (**P**) and (**N**) are the positive and negative controls, respectively.

**Figure 13 nanomaterials-09-00765-f013:**
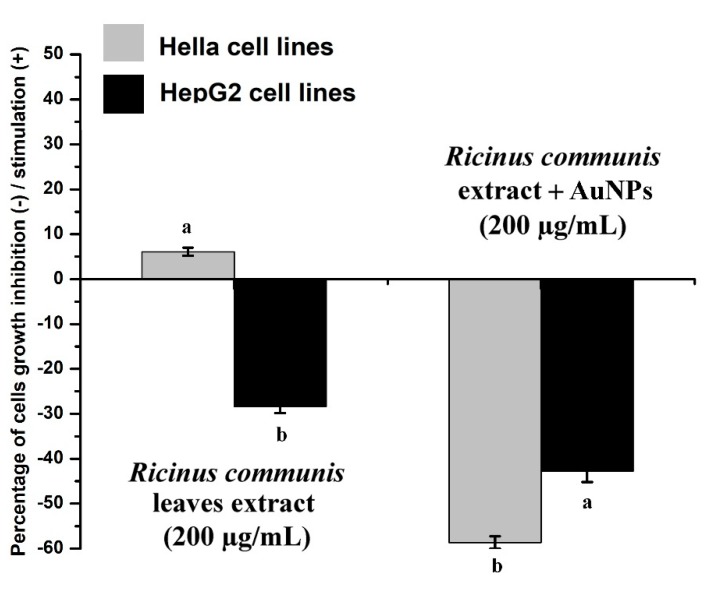
Effect of *Ricinus communis* leaf extract (RcExt) and gold nanoparticles fabricated with extract (RcExt-AuNPs) on Hela and HepG2 cell line growth. All data are represented as mean ± standard deviation (indicated as error bars) from three determinations. Bars with letters in each group indicate a significant (*p* ≤ 0.05) difference among the treatments.

**Table 1 nanomaterials-09-00765-t001:** Chemical composition of *Ricinus communis* leaf extract (RcExt) using GC-MS/MS.

Retention Time (min)	Compounds	Molecular Formula	Molecular Weight
7.13	a, a-Gluco-octonic acid lactone	C_8_H_14_O_8_	238
7.23	Ethyl a-d-glucopyranoside	C_8_H_16_O_6_	208
10.14	Hexadecanoic acid	C_17_H_34_O_2_	270
11.19	3-Pyridinecarbonitrile	C_8_H_8_N_2_O_2_	164
12.75	Phytol	C_20_H_40_O	296
12.97	Methyl stearate	C_19_H_38_O_2_	298
16.01	2,4-Cresotic acid	C_29_H_30_O_10_	538
18.12	2-hydroxy-1-(hydroxymethyl) ethyl ester	C_19_H_38_O_4_	330
20.13	Octadecanoic acid	C_21_H_42_O_4_	358
23.98	Vitamin E	C_29_H_50_O_2_	430
27.41	a-Amyrin	C_30_H_50_O	426
28.24	Lupeol	C_30_H_50_O	426

All compounds identified based upon spectral similarity with the MS library (NIST MS Search 2.0). No chemical reference standards were used.

**Table 2 nanomaterials-09-00765-t002:** Retention time, area, area %, height, and height % of the peaks separated by high-performance liquid chromatography/ultraviolet-visible (HPLC/UV-VIS) data of *Ricinus communis* leaf ethanol extract at the wavelength of 220 nm.

Peak #	Retention Time	Area	Area%	Height	Height%
1	0.458	15,104	0.046	2237	0.141
2	0.625	3741	0.011	536	0.034
3	0.833	64,428	0.196	14,053	0.888
4	1.208	11,160,808	33.974	648,863	40.982
5	1.483	5,687,035	17.311	306,533	19.361
6	2.058	14,276,700	43.459	574,230	36.268
7	3.167	311,400	0.948	12,518	0.791
8	3.633	383,032	1.166	6927	0.438
9	5.042	142,801	0.435	4220	0.267
10	5.892	179,320	0.546	3861	0.244
11	6.975	36,198	0.110	1422	0.090
12	7.458	90,968	0.277	1204	0.076
13	10.500	4713	0.014	121	0.008
14	11.975	8137	0.025	223	0.014
15	16.300	37,681	0.115	794	0.050
16	17.158	106,382	0.324	958	0.061
17	30.433	277,649	0.845	3739	0.236
18	51.867	65,207	0.198	841	0.053
	Totals	32,851,304	100.000	1,583,280	100.000

**Table 3 nanomaterials-09-00765-t003:** Retention time, area, area %, height, and height % of the peaks separated by High-performance liquid chromatography/ultraviolet-visible (HPLC/UV-VIS) data of *Ricinus communis* leaf ethanol extract at the wavelength of 256 nm.

Peak #	Retention Time	Area	Area%	Height	Height%
1	0.583	1418	0.015	129	0.026
2	1.150	1,027,787	11.065	84,294	17.277
3	1.525	3,803,372	40.945	162,206	33.245
4	2.042	2,442,100	26.291	143,116	29.333
5	2.267	1,548,996	16.676	90,022	18.451
6	4.550	50,734	0.546	1955	0.401
7	5.208	28,354	0.305	997	0.204
8	8.725	14,372	0.155	309	0.063
9	10.333	66,916	0.720	1307	0.268
10	15.958	36,085	0.388	1000	0.205
11	18.450	2599	0.028	106	0.022
12	28.150	74,939	0.807	1002	0.205
13	49.525	191,205	2.058	1463	0.300
	Totals	9,288,877	100.000	487,906	100.000

**Table 4 nanomaterials-09-00765-t004:** Retention time, area, area %, height, and height % of the peaks separated by High-performance liquid chromatography/ultraviolet-visible (HPLC/UV-VIS) data of *Ricinus communis* leaf ethanol extract at the wavelength of 300 nm.

Peak #	Retention Time	Area	Area%	Height	Height%
1	0.667	369	0.005	74	0.022
2	1.125	4,410,901	54.582	173,353	51.625
3	2.067	3,612,019	44.697	160,406	47.769
4	4.624	14,444	0.179	905	0.272
5	5.075	43,451	0.538	1057	0.315
	Totals	8,081,184	100.000	335,795	100.000

**Table 5 nanomaterials-09-00765-t005:** The lytic effect of *Ricinus communis* ethanol leaf extract (RcExt) and extract with gold nanoparticles (RcExt-AuNPs) on red blood cells (RBCs).

No.	Treatment	Absorbance at the Wave Length of 576 nm	Hemolysis (%)
1	RcExt	0.136 ± 0.002 ^b^	4.15 ± 0.035 ^a^
2	RcExt-AuNPs	3.00 ± 0.000 ^a^	100 ± 0.000 ^b^
3	Control (Negative)	0.0123 ± 0.00 ^c^	0 ± 0.000 ^c^
4	Control (Positive)	3.00 ± 0.000 ^a^	100 ± 0.000 ^b^

All data are represented as mean ± standard deviation (SD) from three determinations of each experiment. Means with different superscript letters in the same column indicate a significant (*p* ≤ 0.05) difference among the treatments and controls.

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
