# Peer review of "Synthesis of Gold Nanoparticles (AuNPs) Using *Ricinus communis* Leaf Ethanol Extract, Their Characterization, and Biological Applications"

_nanomaterials, 2019, doi:10.3390/nano9050765_

Round 1

Reviewer 1 Report

Accepted in present form

Reviewer 2 Report

The manuscript was moderately revised.

This manuscript is a resubmission of an earlier submission. The following is a list of the peer review reports and author responses from that submission.

Round 1

Reviewer 1 Report

While I am in favor of considering to accept this article for publication, I like to recommend few minor edits;

No significant difference were observe in the UV spectrum. The absorbance peaks are hardly visible. I recommend to include high quality UV spectrum if possible. If not, please take of this spectrum and simply mention the absorbance values.

In most of cases, statistical values values and information are not mentioned. For example, in figure 5, error bars are seen but the it not mentioned if that is SE or SD!  And how that error was calculated? How many time the tests were repeated?

Likewise, for figure 6 and 8; it is recommended to mention the information.

for hemolysis study, how many times the experiment was tested? I believe the experiment should be done at least 3 times, if not more, for publication purposes. If not, please redo the test multiple times and include standard deviation (SD) or standard error (SE).

In the last sentence of conclusion- what does it mean by ".....antimicrobial agents6. Patents "

In the section 4.8, the author mentioned about statistical analysis and P value. However, I did not see any P values in any of the figures!

Author Response

Dear Dr. Rabia Li

Nanomaterials Editorial Office                                                          Dated: 05 March 2019

Assistant Editor

MDPI Branch Office, Beijing

Revised Manuscript ID. nanomaterials-456259 titled: " Synthesis of Gold Nanoparticles (AuNPs) using Ricinus communis Leaf Ethanol Extract, their Characterization, and Biological Applications”

Thank you very for your email on 26 February 2019 regarding revision request for the above mentioned manuscript.

The revised the manuscript is hereby uploaded in the journal online portal. We have accepted almost all the suggestions and comments by the reviewers and responded to each point as required. We will be happy to adopt any further change suggested by the reviewers and editor to improve the quality of paper.

Note: In addition to reviewers comments we have made the following changes in the revised manuscript. 

1.      We have revised the author list.

2.      Acknowledgments are revised

With kind regards

Dr. Khalid A. Khan

(Corresponding author)

RESPONSE TO REVIEWERS COMMENTS

Q.1=  No significant difference was observed in the UV spectrum. The absorbance peaks are hardly visible. I recommend to include high-quality UV spectrum if possible. If not, please take of this spectrum and simply mention the absorbance values.

Response: UV- Vis analysis was repeated. Now there is a significant difference in spectrum of plant extract and extract with gold nanoparticles. Figure. 2 is replaced with high quality along with absorbance values.

Q.2. In most of cases, statistical values and information are not mentioned. For example, in figure 5, error bars are seen but the it not mentioned if that is SE or SD!  And how that error was calculated? How many time the tests were repeated?

Reply: Corrected according to the suggestion. Statistical values and all other information are now properly throughout the manuscript. Fig. 5 now has clear description of error bars, their calculation, and times of repetition of the test. 

Q.3. Likewise, for figure 6 and 8; it is recommended to mention the information.

Reply: Fig. 6 and Fig. 8 are now corrected according to the suggestion. These figures have complete description along with statistics.

Q.4. for hemolysis study, how many times the experiment was tested? I believe the experiment should be done at least 3 times, if not more, for publication purposes. If not, please redo the test multiple times and include standard deviation (SD) or standard error (SE).

Reply: Hemolysis and all other experiments were repeated at three biological replicates. Standard deviation is now included in the Table 1.  

Q.5. In the last sentence of conclusion- what does it mean by "...antimicrobial agents6. Patents "

Reply: "...antimicrobial agents6. Patents " was typing mistake which is now corrected. Actually, in the word template of “nanomaterials” journal, there was a heading after “conclusion”  named as ‘Patents”

Q.6.  In the section 4.8, the author mentioned about statistical analysis and P value. However, I did not see any P values in any of the figures!

Reply: Corrected. All the figures throughout the manuscript now have been properly mentioned about the p values.

Reviewer 2 Report

Dear authors, the paper is interesting but I suggest the publication after the following revisions.

Major

Material and Method section have to be the second one not the fourth;

About section  2.1.1 provide the UV-Vis spectra of the both solution to scientifically support the change in color;

Provide  characterization of Ricinus extract (i.e. HPLC, Total Phenolic, Antioxidant Activity);

Section 3, lines 195-196 "The antimicrobial activity [...] uniform distribution", How you can prove the uniform distribution? In the paper nothing about it it's present. Please provide a DLS or AFM characterization with statistic relevance.

Minor

Change the color of figures 5 and 8;

Format the references 23,32,36;

Misprints in line 331

Author Response

Dear Dr. Rabia Li

Nanomaterials Editorial Office                                                          Dated: 05 March 2019

Assistant Editor

MDPI Branch Office, Beijing

Revised Manuscript ID. nanomaterials-456259 titled: " Synthesis of Gold Nanoparticles (AuNPs) using Ricinus communis Leaf Ethanol Extract, their Characterization, and Biological Applications”

Thank you very for your email on 26 February 2019 regarding revision request for the above mentioned manuscript.

The revised the manuscript is hereby uploaded in the journal online portal. We have accepted almost all the suggestions and comments by the reviewers and responded to each point as required. We will be happy to adopt any further change suggested by the reviewers and editor to improve the quality of paper.

Note: In addition to reviewers comments we have made the following changes in the revised manuscript. 

1.      We have revised the author list.

2.      Acknowledgments are revised

With kind regards

Dr. Khalid A. Khan

(Corresponding author)

RESPONSE TO REVIEWERS COMMENTS

MAJOR COMMENTS:

Q.1. Material and Method section has to be the second one not the fourth;

Reply: Materials and methods section has been shifted to second one place instead of fourth.

Q.2. About section  2.1.1 provide the UV-Vis spectra of the both solution to scientifically support the change in color;

Reply: UV- Vis analysis is repeated. Now there is a significant difference in the spectrum of plant extract and extract with gold nanoparticles (Fig.2) which scientifically support the change in color.

Q.3. Provide  characterization of Ricinus extract (i.e. HPLC, Total Phenolic, Antioxidant Activity);

Reply: It would be good enough to characterize the plant extract. But in this study, we made FTIR to know the type of bioactive molecules qualitatively. So, we think the qualitative analysis is enough and humbly request to the reviewer to waive off this comment, please.

Q.4.  Section 3, lines 195-196 "The antimicrobial activity [...] uniform distribution", How you can prove the uniform distribution? In the paper nothing about it it's present. Please provide a DLS or AFM characterization with statistic relevance.

Reply: Corrected and removed the text regarding “uniform distribution

MINOR COMMENTS:

Q.1. Change the color of figures 5 and 8;

Reply: Figures (5 and 8) are corrected. Now their color is changed

Q.2. Format the references 23,32,36;

Reply: Reference number 23, 32, and 36 are changed. Now all the references in the manuscript are according to journal style. 

Q.3. Misprints in line 331

Reply: Corrected

Round 2

Reviewer 1 Report

The authors have revised the manuscript extensively according to my comments and concerns. The present form of the manuscript looks a sound form and I advise for considering to accept for publication.

Author Response

Dear Dr. Rabia Li

Nanomaterials Editorial Office                                                          Dated: 20 March 2019

Assistant Editor

MDPI Branch Office, Beijing

Revised Manuscript ID. nanomaterials-456259 titled: " Synthesis of Gold Nanoparticles (AuNPs) using Ricinus communis Leaf Ethanol Extract, their Characterization, and Biological Applications”

Thank you very for your email on 08 March 2019 regarding revision request for the above mentioned manuscript.

The revised the manuscript is hereby uploaded in the journal online portal. We have accepted all the suggestions and comments by the reviewers and responded to each point as required. We will be happy to adopt any further change suggested by the reviewers and editor to improve the quality of paper.

With kind regards

Dr. Khalid A. Khan

(Corresponding author)

                                             RESPONSE TO REVIEWERS COMMENTS

"Synthesis of Gold Nanoparticles (AuNPs) using   Ricinus communis Leaf Ethanol Extract, their Characterization, and Biological   Applications”

REVIEWER-1  

Comments

The   authors have revised the manuscript extensively according to my comments and   concerns. The present form of the manuscript looks a sound form and I advise   for considering to accept for publication.

Q.1

English   language and style are fine/minor spell check required

Answer

English language and style is checked   again by Prof. William N. Setzer (co-author) who is a native English speaker.

Reviewer 2 Report

The key characterizations required in the previous revision were not provided. Hence, in my opinion this paper can not be published in the present form. 

The extract caratcherization with FTIR is too poor, other techniques  as HPLC have to be used. For NPs characterization is the same, have to be used a technique that provides the distribuition size of NPs.

Author Response

Dear Dr. Rabia Li

Nanomaterials Editorial Office                                                          Dated: 20 March 2019

Assistant Editor

MDPI Branch Office, Beijing

Revised Manuscript ID. nanomaterials-456259 titled: " Synthesis of Gold Nanoparticles (AuNPs) using Ricinus communis Leaf Ethanol Extract, their Characterization, and Biological Applications”

Thank you very for your email on 08 March 2019 regarding revision request for the above mentioned manuscript.

The revised manuscript is hereby uploaded in the journal online portal. We have accepted all the suggestions and comments by the reviewers and responded to each point as required. We will be happy to adopt any further change suggested by the reviewers and editor to improve the quality of paper.

With kind regards

Dr. Khalid A. Khan

(Corresponding author)

RESPONSE TO REVIEWERS COMMENTS

"Synthesis of Gold Nanoparticles (AuNPs) using   Ricinus communis Leaf Ethanol Extract, their Characterization, and Biological   Applications”

REVIEWER   -2

The key   characterizations required in the previous revision were not provided. Hence,   in my opinion this paper cannot be published in the present form.

Q.1

Moderate   English changes required

Answer

English language and style is checked again by Prof.   William N. Setzer (co-author) who is a native English speaker.

Q.2

The   extract characterization with FTIR is too poor, other techniques  as HPLC have to be used.

Answer

The composition of plant extract is now analyzed by   GC-MS/MS. Its detail is included in the revised text.  HPLC is not available in our department.  

Q.3

For NPs characterization is the same, have to be used a technique   that provides the distribution size of NPs

Answer

An X-ray diffraction (XRD) measurement of the gold nano   particles was recorded and included in the revised manuscript
